# Overburden Failure Associated with Slicing Mining in a Super Thick Coal Seam under Special Weak Aquifers

Kai Chen [1,2,3,*], Ying Ge [1,4], Zhiqi Liu [1], Lifeng Chen [1] and Quan Zhang [1]

1 School of Geology and Mining Engineering, Xinjiang University, Urumqi 830017, China
2 School of Resource and Earth Science, China University of Mining & Technology, Xuzhou 221116, China
3 State Key Laboratory for Geomechanics and Deep Underground Engineering, Xinjiang University, Urumqi 830017, China
4 Discipline of Civil and Infrastructure Engineering, School of Engineering, Royal Melbourne Institute of Technology, RMIT University, Melbourne, VIC 3001, Australia
* Correspondence: chenk412@xju.edu.cn; Tel.: +86-186-9915-7981

**Abstract:** With the increasing improvement of national ecological standards, the eco-environmental problems caused by super thick coal seam mining in western China are becoming more and more serious. The failure law of weak overburden stratum is an important factor affecting the safe mining of coal. The failure characteristics of weakly cemented overburden under high-intensity mining in the mining area of western China were studied. For this purpose, a case study was conducted in the 1101 working face of the Baituyao Coal Mine in Ürümqi County. Based on the analysis of geological conditions in the study area, we combined empirical calculations with engineering analogy, physical simulation, and numerical simulation to comprehensively analyze the characteristics of mining-induced overburden failure. The study showed that the overburden in the study area had several unfavorable engineering geological characteristics, including ease of softening in the presence of water. The Middle Jurassic Xishanyao Formation is a directly recharged aquifer with a weak water-retaining property. Overburden failure mainly occurred at the two ends of the open-off cut. During the mining process, vertical fissures and bed-separated fissures were periodically developed and closed, and the fissures were interconnected. The overburden was fractured, and the fractured zone showed a trapezoidal shape, tapering off from bottom to top. The heights of the caving zone and the water-conducting fracture zone were 25 and 280 m, respectively, in the 1101 working face of the Baituyao Coal Mine, and the ratio of fracturing to mining height was 14.0. Due to the weakly cemented overburden and the presence of the Neogene weak aquifer, the risk of water and sand bursts still exists in this working face under high-intensity mining. Our findings shed light on the safe mining and environmental protection of the ground surface in coal mine shafts in western China.

**Keywords:** super thick coal seam; overburden failure; physical simulation; numerical simulation

## 1. Introduction

Western China is rich in coal resources, which account for 82.8% of China's total coal reserves [1]. The coal resources in western China feature a large mining thickness, shallow burial depth, simple geological conditions, and fast advance of the working face. These coal resources are suitable for high-intensity mining [2,3]. However, western China is an ecologically vulnerable region where coal mining is more likely to result in the loss of surface water and groundwater, thereby threatening surface vegetation and underground mining safety. For mining areas in western China, there is an urgent need to solve the water and sand burst hazards under weakly water-rich conditions while achieving water-preserved-mining [4–6]. Apart from China, other countries along the Silk Roads are also faced with the same engineering problems, including Russia, Mongolia, and Kazakhstan. Concerning slice mining of super thick coal seams under a particularly weak aquifer,

studying the failure height of overburden is highly important for safe mining of the coal mine and ecological environment protection of the land surface.

Theories for safe mining have been established for coal seams in mining areas of eastern China under large water bodies, such as rivers, lakes, and seas [7–10]. Zhang [11] and Hu et al. [12] studied the Beizao Coal Mine near the Bohai Sea and the Cuizhuang Coal Mine in Weishan Lake. The deformation and failure characteristics of overburden under multiseam mining were analyzed by numerical simulation. Sun et al. [13] attempted to resolve the water burst problem associated with mining beneath the Xiaolangdi Reservoir. The process of overburden failure under various mining conditions was characterized. They derived the calculation formulas and subjected them to industrial verification. Feng et al. [14] discussed the safety problems of fully mechanized sub-level caving of super thick coal seams under reservoirs. Zhang et al. [15] analyzed the deformation and failure laws of overburden under coal mining in a deep geological environment. Guo et al. [16] proposed a new method for predicting the height of the water-conducting fracture zone under the above-mentioned mining technique. Dmytro Rudakov et al. [17] presented and validated an analytical model of water inflow and rising level in a flooded mine and examined the model robustness and sensitivity to variations of input data considering the examples of three closed hard-coal mines in Germany. Bazaluk, O. et al. [18] analyzed the dynamics of the underground water of a mine field based on the study of the geofiltration process of the rock mass disturbed by mining to achieve safe extraction operations as well as subsurface territories at the stage of the mining enterprise closure.

Western China is rich in coal resources and ecologically vulnerable. The coal seams usually have shallow burial depths and large mining thicknesses. These coal seams are also influenced by fissure water in Jurassic sandstones, surface water, and loose rock pore aquifers. Many mining areas face dual challenges: preventing mine water disasters and carrying out water conserving mining operations simultaneously [19–21]. Chen et al. [22] analyzed the fractal evolution law of mining-induced overburden fractures in super thick coal mines by combining fractal geometry theory with discrete element numerical simulation. Qiao [23] investigated the overburden failure characteristics in fully mechanized caving with a significant mining height and modified the conventional empirical formulas. Han et al. [24] analyzed the overburden failure characteristics under sublevel fully mechanized caving mining for super thick coal seams with weak overburden. They further determined the relationship between the extraction thickness and overburden failure height. Zeng et al. [25], Qin et al. [26], and Guan et al. [27] studied the overburden structural evolution and ground pressure control under slice mining of super thick coal seams based on the stratigraphic column chart and lithology of the Dajing mining area in Xinjiang's Zhundong Coalfield.

A lot of research has been done on the evolution law of mining overburden fractures under large water bodies, such as rivers, lakes, and seas in the existing research results. The coal mining conditions and hydrogeological structure characteristics in western China's mining areas are quite different from those in eastern China's coal fields. Under such special weak aquifer conditions, there are some problems, such as how the overburden failure law of shallow buried thick coal seam mining is, and what are the characteristics of the height development of water conducting fracture zones, All need further research, discussion, and verification. At the same time, the "one belt—one road" initiative has been continuously strengthened, and coal resources are constantly exploited in western China. In this context, slice mining of super thick coal seams is associated with the prevalence of water and sand burst disasters when the aquifers have low water-richness. To ensure safe mining and water resource protection of mining areas in western China, it is necessary to clarify overburden failure characteristics under the mining of super thick coal seams. The 1101 working face of the Baituyao Coal Mine in Ürümqi County was the object of study in this paper. We first analyzed the typical coal seam occurrence conditions and the hydrological and geological features in this working face. We combined a physical analog model with numerical simulation to analyze the overburden failure characteristics under

slice mining of super thick coal seams with a special weak aquifer. The research findings provide scientific evidence and a technical reference for the safe mining of super thick coal seams and water resource conservation in mining areas of western China.

## 2. Materials and Methods

### 2.1. Geological Conditions

The Baituyao Coal Mine, situated on the south margin of the Junggar Basin in the Xinjiang Uygur Autonomous Region, China (Figure 1), covers an area of 6.65 km² and has an annual coal output of 1.2 million tons. The mine field is divided into two levels for mining, with a horizontal elevation of +600 m and +200 m. The strike length of the working face 1101 is 1000 m, and the dip length is 154 m. The fully mechanized sub-level caving method is adopted. The study area is hilly and has the Gobi desert, lying higher in the west and lower in the east. The ground elevation varies between 1050 and 1182 m, with a relative height difference of 132 m. Only a few gullies are developed on the ground surface. The coalfield extends from the southwest to the northeast, dips to the northwest, and has a dip angle of 25–30°. It has a monoclinal structure, with only the reverse fault F4 developed in the southeastern part of the study area. There is no magma intrusion in the study area. Only a few bedrock outcrops are observed in the western part of the study area. Significant parts of the study area are covered by Quaternary overburden.

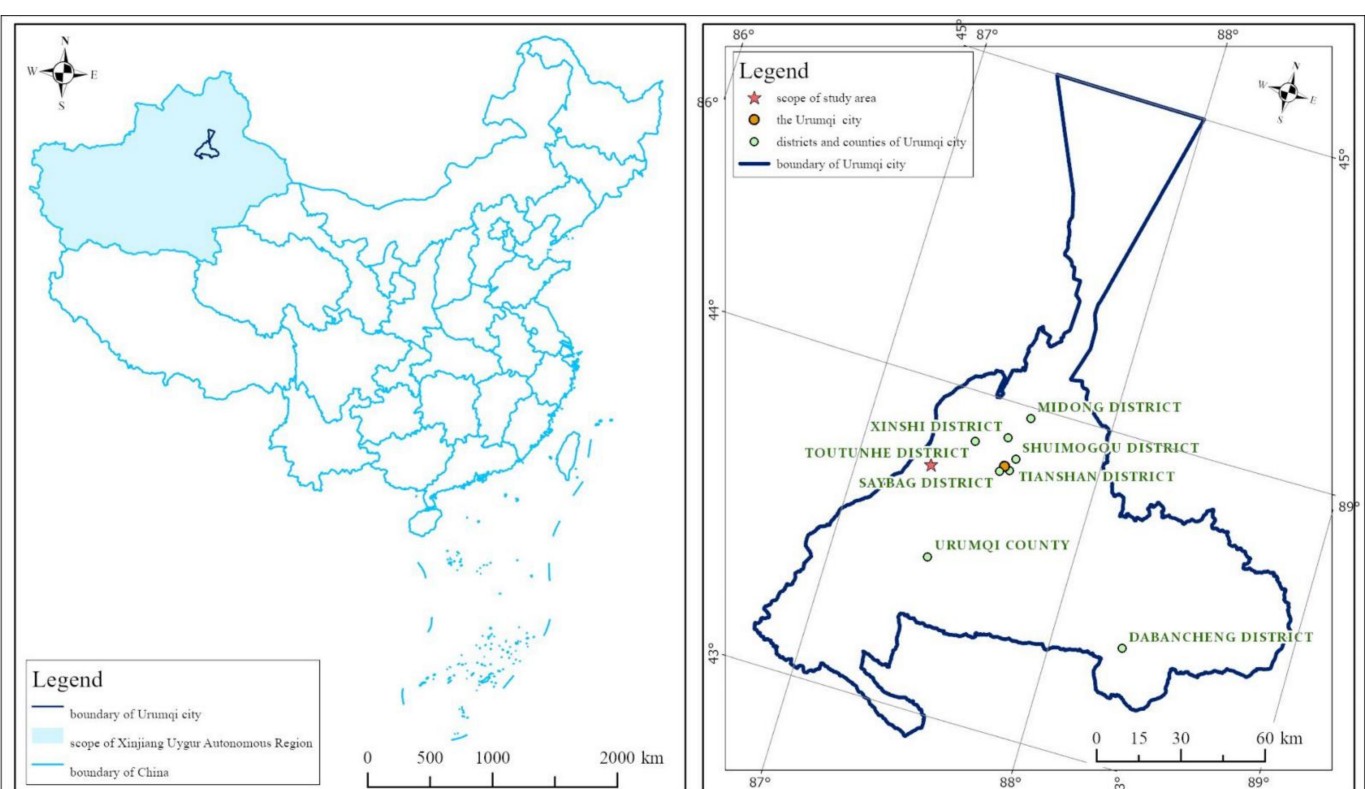

**Figure 1.** Location of the Sima Coal Mine.

Two production coal seams in the Baituyao Coal Mine (coal seams B1 and B2) are stable and recoverable throughout. Here, we mainly discuss coal seam B1 in the 1101 working face. Coal seam B1 is the main mining seam, with a thickness of 17 to 23.5 m and a dip angle of approximately 22°. There are six aquifers (aquicludes) in the study area (Table 1).

**Table 1.** Division of aquifers (aquicludes).

| Formation Code | No. | Aquifer (Aquiclude) | Water Abundance |
|---|---|---|---|
| Q | I | Quaternary permeable, nonretaining layer | Weak to intermediate |
| $N_2d$ | II | Porous aquifer of Pliocene Dushanzi Formation | Good |
| $J_2t$ | III | Aquiclude of Middle Jurassic Toutunhe Formation | |
| $J_2x$ | IV | Pore-fissure aquifer of Middle Jurassic Xishanyao Formation | Weak |
| $J_1s$ | V | Aquiclude of the Middle Jurassic Toutunhe Formation | |
| | VI | Fissure unconfined aquifer composed of burnt rocks | |

The coal seam has a shallower burial depth near the outcrops than elsewhere, which damages the locally impermeable layer. As a result, the groundwater in the Neogene aquifer recharges the pore-fissure water-retaining aquifer of the Middle Jurassic Xishanyao Formation or directly flows into the roadways. In addition, the fissure unconfined aquifer composed of burnt rocks is recharged by atmospheric precipitation and melt water, which then becomes unconfined water in the fissures of burnt rocks. Such water directly recharges the pore-fissure aquifer of the Middle Jurassic Xinshanyao Formation, giving rise to hydraulic connections between aquifers.

The thickness of the Quaternary loose soils overlying the coal-measure strata in the study area varies between 0 m and 71.73 m, with a mild decreasing trend from south to north. The lithologies include sandy loam, sand, loess, and gravel. The overburden thickness is 0–250 m in the coal seams of the study area. The overall overburden thickness is small, and the thickness may be slight at specific positions. The lithologies include Neogene and Jurassic conglomerate, mudstone, sandstone, silty mudstone, and argillaceous siltstone, as shown in Figure 2.

| | System | Formation | Depth | Thickness | Aquiclude | Hydrogeological characteristics |
|---|---|---|---|---|---|---|
| | Quaternary | | 46.51 | 46.51 | Permeable aquifer | This section is mainly Quaternary overburden, and its lithology is mainly sandy soil. |
| | Neogene | Dushanzi Formation of Pliocene($N_2d$) | 126.25 | 79.74 | Neogene Pliocene Dushanzi formation pore fissure weak medium water rich aquifer | The lithology of the upper and middle parts of this section is mainly grayish white, gray mudstone and silty mudstone. Only the bottom part is developed with sandy conglomerate, and the pores and fissures are relatively developed. It is a weak medium water rich aquifer. |
| | Jurassic | Middle Toutunhe Formation($J_2t$) | 198.81 | 72.56 | Middle Jurassic Toutunhe Formation aquifer | The lithology of this section is mainly purplish red and gray mudstone, with thin layers of argillaceous cemented coarse sandstone and medium sandstone locally developed. The water yield is very weak and it is a water resisting layer. |
| | | Middle Xishanyao Formation($J_2x$) | | | Middle Jurassic Xishanyao Formation Pore fissure weak water rich aquifer | The lithology of this section is mainly coarse sandstone, medium sandstone, fine sandstone, siltstone, argillaceous siltstone, silty mudstone, mudstone, carbonaceous mudstone, etc., which are interbedded with coal seams B1 and B2. According to the principle of water bearing and water resisting layer division, coarse fine sandstone is the aquifer, argillaceous siltstone, silty mudstone, mudstone, carbonaceous mudstone and coal are the aquifers, and the water yield is very weak, so it is a weak water rich aquifer. |
| | | coal seams B2 | | | | Located in the lower part of the coal bearing stratum, the coal seam is 8.27 ~ 23.50m thick, with an average thickness of 15.47m. The lithology of the roof of the coal seam is mainly mudstone and siltstone, and the lithology of the floor is clay rock. It is 9.96 ~ 18.01m away from the B1 coal seam, with an average spacing of 14.78m. |
| | | coal seams B1 | | | | Located in the lower part of the coal bearing stratum, the coal seam is 8.27 ~ 23.50m thick, with an average thickness of 15.47m. The lithology of the roof of the coal seam is mainly mudstone and siltstone, and the lithology of the floor is clay rock. It is 9.96 ~ 18.01m away from the B1 coal seam, with an average spacing of 14.78m. |
| | | | 505.13 | 220.31 | | |

**Figure 2.** The lithology of the site under study.

Compared with the Carboniferous coal-bearing strata in eastern China, the Neogene and Jurassic strata in the study area share some common features. For example, short diagenetic history, low cementation, and ease of softening in the presence of water. The

lithology and mechanical properties of the floor and roof of the 1101 working face are shown in Table 2.

**Table 2.** The lithology and mechanical properties of the floor and roof of the 1101 working face.

| Rock Type | Lithology | Uniaxial Compressive Strength (MPa) | | Tensile Strength (MPa) | Shear Strength (MPa) | Particle Density (g/m³) | Assessment Result | Notes |
|---|---|---|---|---|---|---|---|---|
| | | Dry | Water-Saturated | Water-Saturated | Water-Saturated | | | |
| B2 roof | Mudstone, argillaceous siltstone | 8.74–32.40 17.91 | 30.80–91.60 45.76 | 0.13–2.15 0.69 | 1.06–6.05 3.32 | 2.37–2.91 2.63 | Soft-relatively hard rock | Min–Max Average |
| B1 roof | Mudstone, argillaceous siltstone | 0.30–18.20 7.94 | 18.08–60.40 36.93 | 0.16–3.44 1.70 | 1.23–11.00 6.55 | 2.40–3.01 2.60 | Extremely soft-relative hard rock | |

Moreover, there is a quicksand layer approximately 8 m thick at the bottom of the Neogene strata, which has an angular unconformable contact with the underlying strata. This working face is threatened by water and sand bursts due to the presence of weakly cemented overburden and Neogene aquifers under the mining of super thick coal seams. Therefore, assessing the risks of overburden failure and sand burst is necessary to determine the feasibility of safe mining.

### 2.2. Empirical Calculation

The statistical, empirical formula recommended by the State Administration of Work Safety is the most widely used method for estimating the height of the water-conducting fracture zone in China [28]. This empirical formula, derived statistically from a large number of field measurements, can satisfy the design requirements for coal mining under water bodies in China to a certain extent. However, the empirical formula only applies to the situation where the mining thickness of a single slice is 1–3 m, and the cumulative mining thickness does not exceed 15 m. The formula may be defective when used to estimate the failure height of overburden under slice mining of super thick coal seams. In the present study, the calculation was conducted under a cumulative mining thickness of 15 m. The overburden in the study area is composed of weak strata. Therefore, the heights of the caving zones and the water-conducting fracture zones in the overburden were calculated using the following empirical formulas:

$$H_m = \frac{100\sum M}{6.2\sum M + 32} \pm 1.5 \tag{1}$$

$$H_{li} = 10\sqrt{\sum M} + 5 \tag{2}$$

where $H_m$ is the height of the caving zone, $m$; $H_{li}$ is the height of the water-conducting fracture zone, $m$; and $\sum M$ is the cumulative mining thickness, $m$.

### 2.3. Physical Analog Model

A physical analog model is one of the commonly used methods to obtain the height of the water-conducting fracture zone in the overburden. Based on a similar theory and the conceptualized engineering geological model, we built a similar material model along the strike of the coal seam (Figure 3).

An analog physical model was designed following the similarity criteria in geometry, time, volumetric weight, and strength. The model had a specification of 2.5 m × 0.3 m × 1.25 m (length × width × height). The geometric factor was 1:240; the time similarity constant was 15.5; the volumetric weight ratio was 1.5, and the strength was 360 kPa. An analog physical model was prepared using ordinary river sand, gypsum, and barytes. The slices were separated from each other by sprinkling mica powder. One 120 m coal pillar was retained on each model side to reduce boundary effects. The extraction length in the middle was 360 m. The coal seam thickness was 20 m, and downward slicing mining was

conducted using the following scheme. Each coal seam was extracted in two slices, and the extraction thickness of each slice was 5 cm (10 cm in the prototype). The open-off cut was located on the model's right side, and the working face was advanced by 10 cm at each mining step (24 m in the prototype), as shown in Figures 4 and 5.

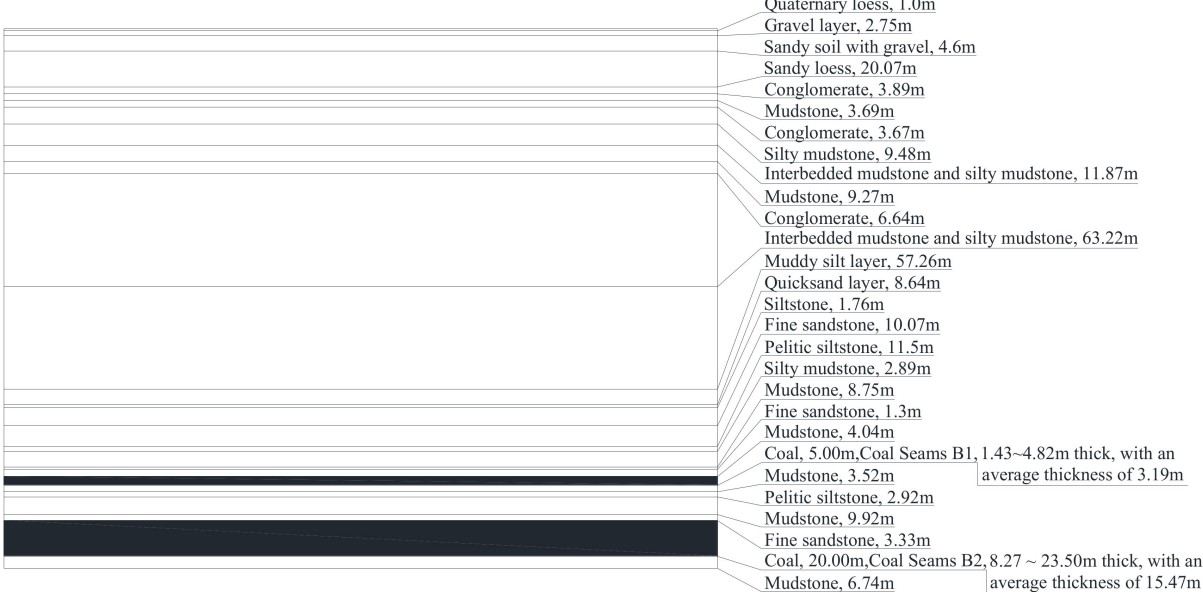

Quaternary loess, 1.0m
Gravel layer, 2.75m
Sandy soil with gravel, 4.6m
Sandy loess, 20.07m
Conglomerate, 3.89m
Mudstone, 3.69m
Conglomerate, 3.67m
Silty mudstone, 9.48m
Interbedded mudstone and silty mudstone, 11.87m
Mudstone, 9.27m
Conglomerate, 6.64m
Interbedded mudstone and silty mudstone, 63.22m
Muddy silt layer, 57.26m
Quicksand layer, 8.64m
Siltstone, 1.76m
Fine sandstone, 10.07m
Pelitic siltstone, 11.5m
Silty mudstone, 2.89m
Mudstone, 8.75m
Fine sandstone, 1.3m
Mudstone, 4.04m
Coal, 5.00m, Coal Seams B1, 1.43~4.82m thick, with an average thickness of 3.19m
Mudstone, 3.52m
Pelitic siltstone, 2.92m
Mudstone, 9.92m
Fine sandstone, 3.33m
Coal, 20.00m, Coal Seams B2, 8.27 ~ 23.50m thick, with an average thickness of 15.47m
Mudstone, 6.74m

**Figure 3.** Physical analog model of the 1101 working face in the Baituyao Coal Mine.

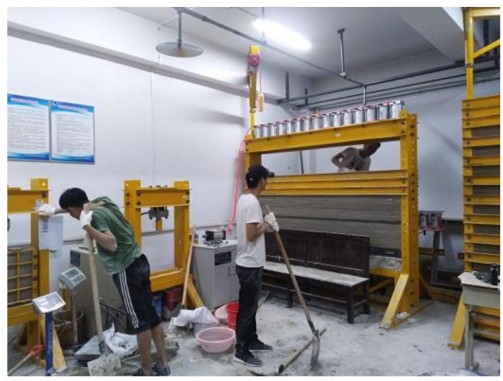

**Figure 4.** Laying process of the physical analog model.

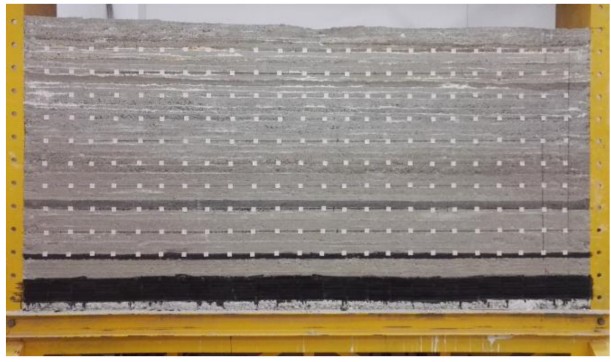

**Figure 5.** Physical analog model.

### 2.4. Numerical Simulation

The Universal Distinct Element Code (UDEC) is a numerical program for discontinuous structures, offering an effective tool for studying the height of water-conducting fracture zones under mining conditions. According to the mining and geological conditions of the 1101 working face in the Baituyao Coal Mine, the base rock overlying the coal seam is approximately 240 m thick. The base rock is overlain by Quaternary alluvium. The recoverable coal thickness of coal seam B1 is 8.27 m to 23.50 m. The average recoverable coal thickness is 15.47 m. The coal seams are recoverable throughout the study area, as the coal seam structure is simple. The coal seam roof consists of mudstone and argillaceous siltstone and quickly softens in the presence of water. The roof has poor water resistance, and the rocks are highly soft to soft. A similar material model was built along the strike of the coal seam, extending horizontally for 600 m and vertically for a height of 297.8 m until reaching the ground surface. One 120 m coal pillar was retained on each model side to reduce the boundary effect. Displacement constraints were imposed on the left and right sides and at the bottom of the model. The upper surface of the model was a free boundary. A schematic diagram of the numerical model is presented in Figure 6.

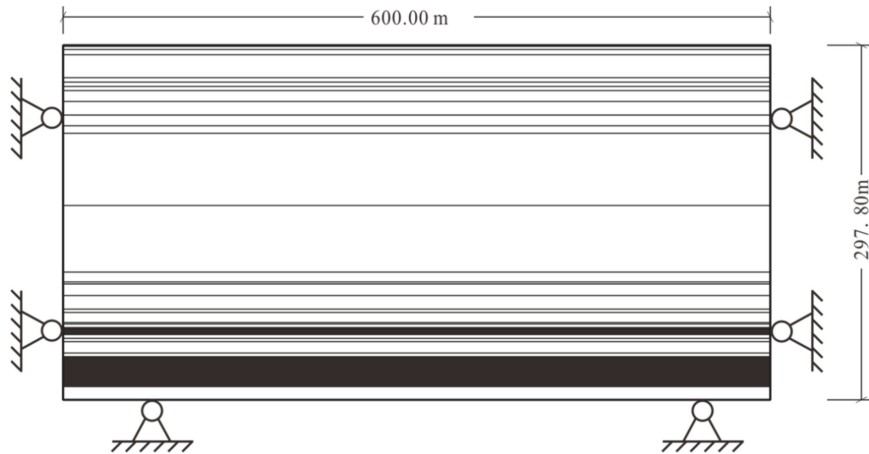

**Figure 6.** Schematic diagram of the numerical model [22].

Physical and mechanical parameters of each stratum are shown in Table 3. The mining scheme and parameters in the numerical simulation were consistent with those of the physical simulation.

**Table 3.** Mechanical parameters of coal and rock strata in the numerical simulation.

| Lithology Code | Elastic Modulus (MPa) | Poisson's Ratio | Cohesion (MPa) | Friction Angle (°) |
|---|---|---|---|---|
| Conglomerate | $5.0 \times 10^3$ | 0.23 | 4.00 | 43.0 |
| Fine sandstone | $2.2 \times 10^3$ | 0.21 | 0.56 | 42.7 |
| Siltstone | $2.8 \times 10^3$ | 0.22 | 1.22 | 37.6 |
| Mudstone | $3.0 \times 10^3$ | 0.38 | 1.67 | 34.4 |
| Coal | $3.1 \times 10^3$ | 0.26 | 1.55 | 35.8 |

## 3. Results

### 3.1. Calculation of the Overburden Failure Height

The roofs of coal seams B1 and B2 in the study area are primarily composed of mudstone and locally of argillaceous siltstone, with argillaceous cementation. According to the physical and mechanical parameters obtained for roof rocks by field drilling, the average uniaxial compressive strength of roof rocks of coal seam B1 was 36.93 and 7.94 MPa in the dry and saturated states, respectively. The average uniaxial compressive strength

of roof rocks of coal seam B2 was 45.76 and 17.91 MPa in the dry and saturated states, respectively. Due to the presence of the Neogene aquifer, this working face was locally threatened by water and sand bursts. Thus, the average uniaxial compressive strength of roof rocks of coal seams B1 and B2 in the saturated state was taken as 12.93 MPa for the calculation.

It is noteworthy that empirical formulas recommended in the specifications for coal mining under buildings, water bodies, and railways only apply to the situation where the mining thickness of a single slice is 1–3 m, and the cumulative mining thickness does not exceed 15 m. The height of the water-conducting fracture zone will be 43.7 m, and the height of the caving zone will be 13.5 m if calculated by a cumulative mining thickness of 15 m. However, the empirical formulas recommended in the specifications for coal mining under buildings, water bodies, and railways have some intrinsic limitations when used to calculate the failure height of overburden under the mining of thick and super thick coal seams. According to the field measurement data for super thick coal seam B1 in the #2 coal mine of the Xinjiang Zhundong Coalfield, which shares similar mining and geological conditions, the height of the water-conducting fracture zone is 145.8 m, and the ratio of fracturing to mining height is 6.1 when the mining thickness is 24 m [24]. Therefore, the height of the water-conducting fracture zone, calculated by reference to the experience in other mining areas of Xinjiang, is 122 m for the fracturing height-to-mining thickness ratio of 6.1

*3.2. Propagation of Overburden Failure under High-Intensity Mining of Coal Seams under Special Weak Aquifers: Physical Analog Model*

As shown by the overburden failure characteristics in Figure 7a–d, the water-conducting fracture zone gradually formed as mining proceeded in the working face.

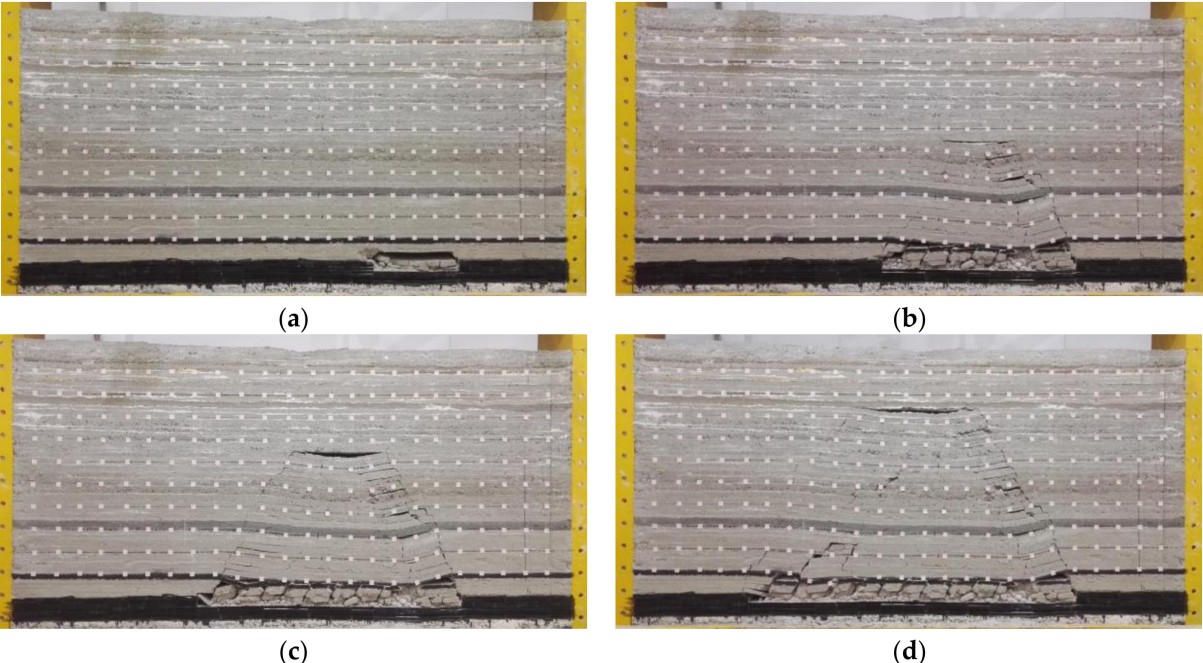

**Figure 7.** Overburden failure characteristics: physical analog model with the working face advancing at 96 m (**a**), 216 m (**b**), 288 m (**c**), and 360 m (**d**).

The overburden remained stable as the coal seam was excavated for a distance of 72 m. Only a small amount of thin-bedded fine-grained sandstones fell off. When the working face advanced to 96 m, the roof collapsed for the first time, forming an inverted step-shaped fractured zone near the open-off cut. As the working face continued to advance, the height of the water-conducting fracture zone continued to increase stably. The fissures

propagated rapidly upward, and the mining-induced fissures formed an inverted V-pattern. The periodic roof weighting interval was approximately 15 to 20 m. The mining stopped after the working face advanced 360 m. At this time, the horizontal fissures increased in number in the overburden. However, in the strata near the goaf, the fissures were gradually closed due to squeezing from the upper collapsed strata and the support provided by the lower fractured rock masses. Vertical fissures developed continuously, resulting in a trapezoidal fractured zone that tapered off from bottom to top. The maximum height of the water-conducting fracture zone was 225.6 m, and the corresponding ratio of fracturing to mining height was 22.56.

### 3.3. Stress, Strain, Displacement, and Overburden Failure from Numerical Simulation

- Failure zone

As shown by the numerical simulation of overburden failure induced by extraction of the first slice from Figure 8a–d, the region of overburden failure gradually increased as the working face advanced continuously.

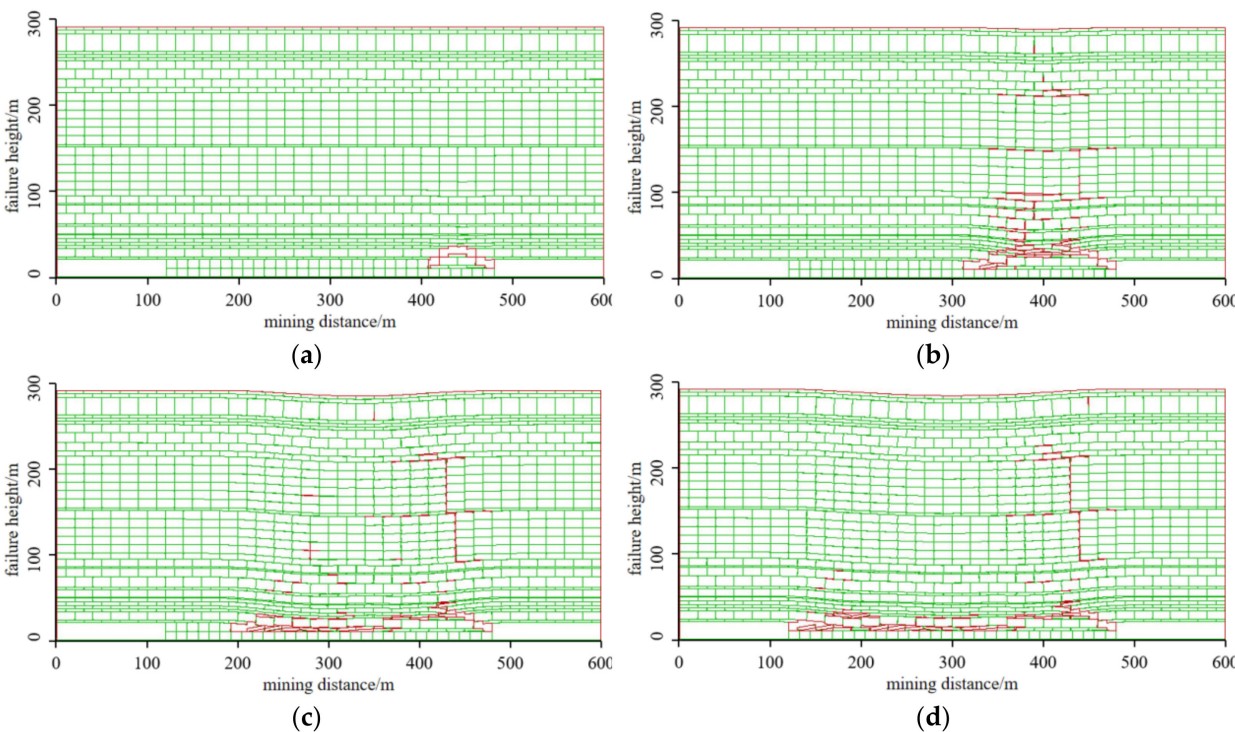

**Figure 8.** Fracture distribution of overburden under the extraction of the first slice with the working face advancing to: (**a**) 72 m (the first slice); (**b**) 168 m (the second slice); (**c**) 288 m (the first slice); (**d**) 360 m (the end of the extraction of the first slice).

Fissures induced in the overburden were divided into vertical and interlayer fissures (tensile fissures extending horizontally). The former was mainly found on the two ends of the open-off cut, causing more significant failure in height and width than in the middle of the overburden. The latter was mainly found in the middle of the overburden, and the bed-separated fissures showed periodic closure as the working face continued to advance. When the working face advanced to 168 m, a mild collapse of the ground surface occurred. The height of overburden failure reached 100 m, and the ratio of fracturing to mining height was 10.0. As the working face continued to advance, fissures were incessantly developed in the overburden, and the vertical fissures increased in height. In contrast, horizontal fissures showed periodic development-closure. When the working face advanced to 360 m, that is, at the end of the extraction of the first slice, the ratio of fracturing to mining height was

21.0. In the physical simulation, the overburden failure height was 225.6 m, and the ratio of fracturing to mining height was 22.6 at the end of the extraction of the first slice. The difference between the results from physical and numerical simulations was 15.6 m. Taking a more conservative stance, the error would be only 6.9% if the result from the physical simulation was the recommended height. This finding also demonstrated the validity of the numerical simulation. The numerical simulation results for the second slice excavation are depicted in Figure 9a–d.

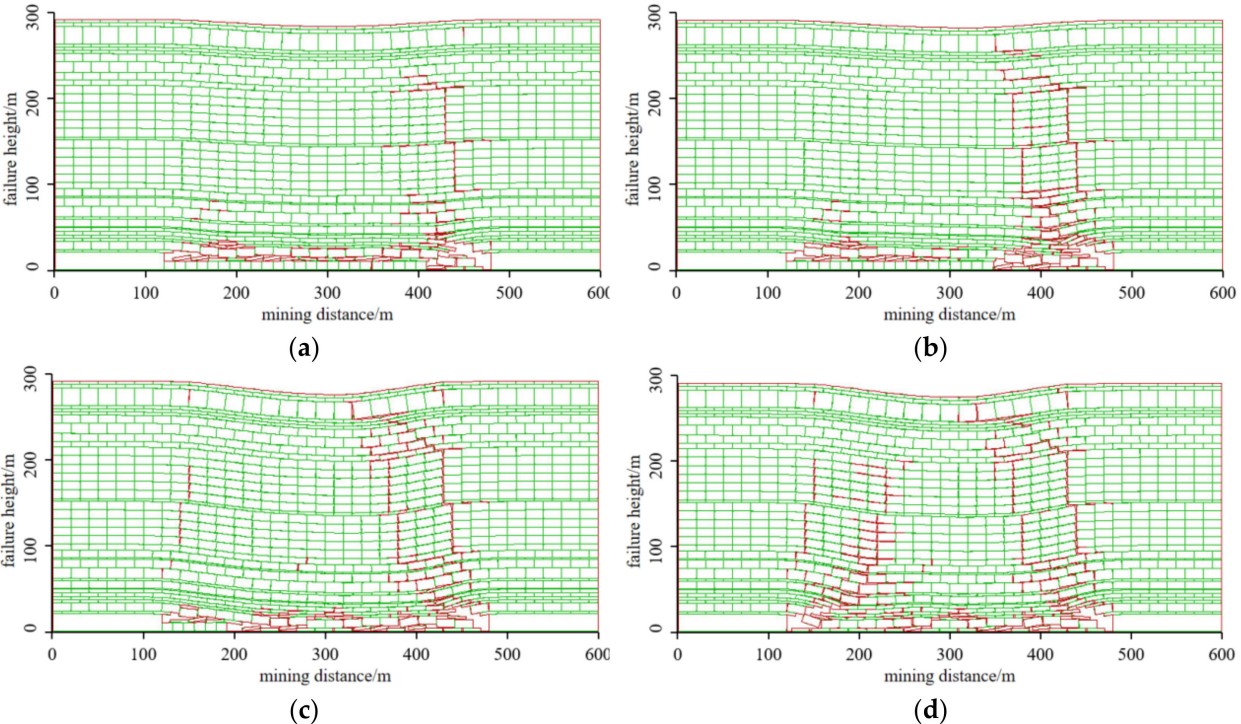

**Figure 9.** Fracture distribution of overburden under the extraction of the second slice with the working face advancing to: (**a**) 72 m (the second slice); (**b**) 132 m (the second slice); (**c**) 288 m (the second slice); (**d**) 360 m (the end of the extraction of the second slice).

The height and width of fissures and the region of overburden failure increased continuously as the extraction of the second slice proceeded. Overburden failure mainly occurred at the two ends of the open-off cut. Vertical fissures and bed-separated fissures were periodically developed and closed. The ground surface also collapsed substantially. Due to the high-intensity mining of the second slice, the region of overburden failure far exceeded that at the end of the extraction of the first slice. In addition, the overburden failure characteristics were consistent between the numerical and physical simulations. A trapezoidal fractured zone tapering off from bottom to top was formed. Under high-intensity mining, fissures occurring at the two ends of the open-off cut were interconnected, causing disturbance to the overlying Neogene and Jurassic weak aquifers. Therefore, the working face was threatened by water and sand bursts. At the end of the extraction of the second slice, the overburden failure height reached 280 m, and the ratio of fracturing to mining height was 14.0.

- Stress zone

A vertical stress from the numerical simulation of overburden failure during extraction of the first slice is shown in Figure 10a,b.

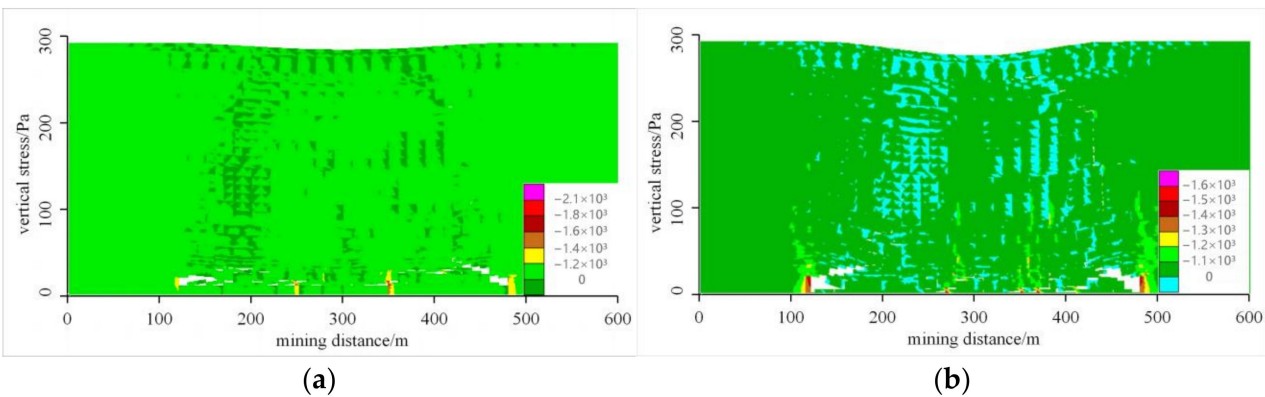

**Figure 10.** Vertical stress of the overburden under mining at the end of the first (**a**) and second (**b**) slice extraction.

During this process, the initial stress field presented a strip-like distribution. At the end of the extraction of the first slice, the vertical stress in the overburden still presented a roughly strip-like distribution. However, a region of increased stress appeared in the upper part of the mining area, where the peak stress reached 5 MPa. As the working face advanced continuously, the initial in situ stress balance could no longer be maintained, leading to stress redistribution. Bed separation occurred due to overburden failure. The stress above the goaf was approximately zero. However, zero-stress regions appeared on the two sides of the goaf. When the working face advanced to 120 m, approximately symmetrical arc-shaped stress concentration regions appeared on the two sides of the goaf, where the peak stress reached 10 MPa. As the upper overburden continued to collapse, the gaps created by bed separation above the goaf were filled, and a region of stress concentration appeared above the goaf. The stress concentration region propagated to another position along the strike along with periodic weighting.

The vertical stress nephogram from the numerical simulation of overburden failure during the extraction of the second slice is shown in Figure 11b. It can be seen that fracturing bed separation occurred again in the overburden. The stress concentration regions gradually disappeared above the goaf, decreasing stress. Like the mining-induced stress variation during the extraction of the first slice, the upper overburden continued to collapse. Several regions of stress concentration appeared above the goaf. At the end of the extraction, the model was completely damaged. The upper strata overlying the coal seam collapsed, and bed separation was observed. Therefore, large parts of the upper overburden were in a stress-free condition. The overburden above the goaf became compact, and a slight stress concentration region appeared here, with the peak stress reaching 20 MPa.

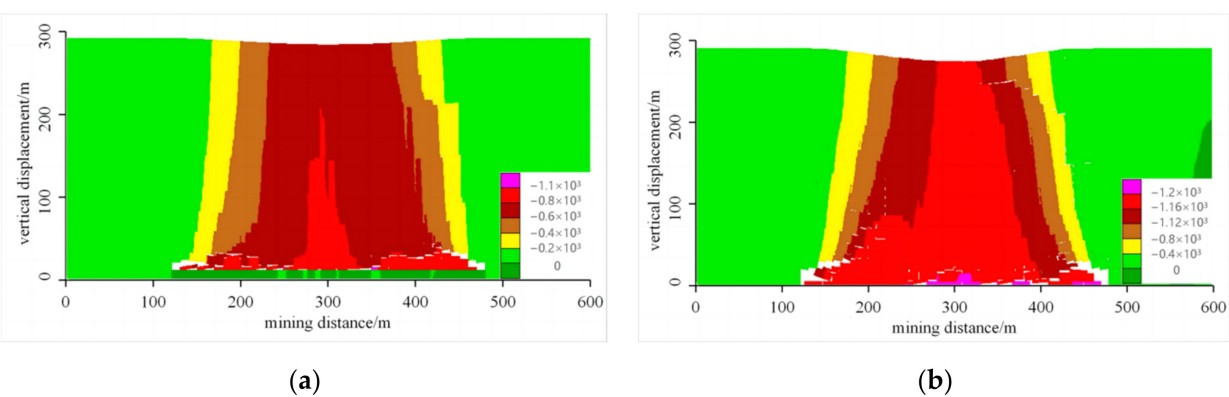

**Figure 11.** Vertical displacement under overburden under mining at the end of the first (**a**) and second (**b**) slice extraction.

- Displacement zone

The vertical stress nephogram from the numerical simulation of overburden failure during the extraction of the first slice is shown in Figure 11a. It can be seen from the figure that as the working face advanced during the extraction of the first slice, the region of deformation expanded gradually. When the working face advanced to 60 m (the first slice extraction), the immediate roof began to collapse, but no deformation was observed in other parts of the overburden. Deformation propagated to the ground surface when the working face advanced to 144 m (the first slice extraction). The rock strata near the ground surface were less significantly deformed. At the end of the extraction of the first slice, all strata were deformed above the goaf. The deformation region presented an inverted V-shape, and the maximum displacement was 8 m.

The vertical stress from the numerical simulation of overburden failure during the extraction of the second slice is shown in Figure 11b. It can be seen that the upper overburden deformation was more significant farther away from the goaf as the extraction of the second slice proceeded and smaller closer to the goaf. The deformation was comparable across the overburden strata. The same variation was observed as the working face advanced during the extraction of the second slice as in the extraction of the first slice. That is, the overburden displacement was smaller farther from the goaf. When the working face advanced to 180 m, the maximum deformation of 16 m occurred in the upper overburden. When the working face advanced to 288 m, the maximum ground surface deformation reached a maximum of 16 m. The ground surface collapsed substantially, resulting in saddle-shaped surface boundaries.

## 4. Discussion

### 4.1. Heights of the Caving Zone and the Water-Conducting Fracture Zone under Mining of an Super Thick Coal Seam

A comparison of the results from the empirical formulas, engineering analogy, physical simulation, and numerical simulation is presented in Table 4.

**Table 4.** Heights of the caving zone and the water-conducting fracture zone.

| Calculation Method | Mining Height (m) | Caving Zone | | Water-Conducting Fracture Zone | |
|---|---|---|---|---|---|
| | | Height (m) | Caving Ratio Mining Height | Height (m) | Caving Ratio Fracturing to Mining Height |
| Empirical formula | 15 | 13.5 | 0.9 | 43.7 | 2.9 |
| Engineering analogy | 20 | / | / | 122 | 6.1 |
| Physical simulation | 10 | 22.19 | 2.2 | 225.6 | 22.6 |
| Numerical simulation | 20 | 25 | 1.3 | 280 | 14.0 |

Taking a conservative assessment, numerical simulation results would prevail when retaining coal (rock) pillars. Due to the presence of weakly cemented overburden and Neogene weak aquifers, the risk of water and sand bursts exists in this working face under high-intensity mining.

It is noteworthy that the empirical formulas recommended in the specifications for coal mining under buildings, water bodies, and railways only apply to the situation where the mining thickness of a single slice is 1–3 m, and the cumulative mining thickness does not exceed 15 m. As mentioned above, the empirical formulas recommended in the specifications for coal mining under buildings, water bodies, and railways have some defects when used to calculate the failure height of overburden under the mining of thick and super thick coal seams. The height of the water-conducting fracture zone under the mining of the super thick coal seam is considerably influenced by lithology. Specifically, the height of the water-conducting fracture zone with soft overburden is 6–11 times the mining thickness, while that of the water-conducting fracture zone with medium hard overburden is 10–20 times the mining thickness. The height of the water-conducting fracture zone with

hard overburden is 16 to 36 times the mining thickness [1]. During calculating the heights of the caving zone and water-conducting fracture zone under super thick coal seam mining, the equations for the heights of the two zones should be fitted and optimized based on the field measurements of the mining area. In addition, other techniques should also be introduced for a comprehensive judgment of the heights of the two zones to achieve a more reliable calculation.

### 4.2. Failure Characteristics of Overburden under the Mining of an Super Thick Coal Seam

The coal resources in western China are characterized by large mining thickness, shallow burial depth, simple geological conditions, and fast advance of the working face [2]. As shown by our physical and numerical simulations, the overburden failure characteristics under the slicing mining of super thick coal seam included the following: (1) The overburden was severely damaged, accompanied by significant ground surface displacement and abnormally large heights of the caving zone and the water-conducting fracture zone; (2) Overburden fissures were mainly concentrated at the two ends of the open-off cut. The vertical fissures and the interlayer fissures were periodically opened and closed. The fissures became interconnected and propagated as the working face advanced at a specific stride length. (3) The stress was divided into shear and tensile stresses within the overburden. The former was the primary stress causing failure, resulting in "cut off" or "shear" failure. The failure characteristics of overburden under the mining of a super thick coal seam obtained by numerical simulation are shown in Figure 12.

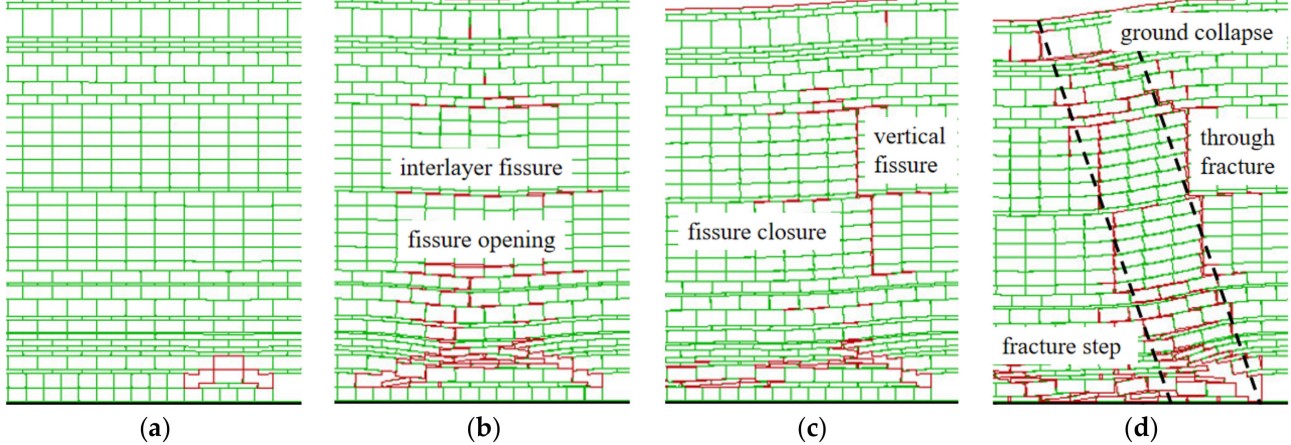

|  (a)  |  (b)  |  (c)  |  (d)  |

**Figure 12.** Failure characteristics of overburden under the mining of a super thick coal seam, obtained by numerical simulation: (**a**) initial collapse; (**b**) periodic caving with fissure opening; (**c**) periodic caving with fissure closure; (**d**) fracture distribution after the caving stabilization.

Due to the weakly cemented overburden and the slicing mining of super thick coal seams in western China, the water-conducting fracture zone presented a trapezoidal shape, tapering from the bottom to the top. In contrast, the fractured zone in overburden in the mining areas of Eastern China had a saddle shape. One reason is that the coal measure strata in the mining area of western China can be dated back to the Cretaceous or Jurassic Period. The overburden of coal seams in the study area features a short diagenetic history, a low degree of cementation, and ease of softening in the presence of water. In addition, coal seams in the mining areas of western China usually have shallower burial depths and larger mining thicknesses. These features are compounded by the unique hydrological and geological features and distinctive engineering geology of the study area. The risk of water and sand burst lingers when the overburden is damaged abnormally or has unfavorable water-retaining properties.

## 5. Conclusions

In this study, empirical calculations, engineering analogy, physical simulation, and numerical simulation were used to comprehensively analyze the characteristics of mining-induced overburden failure. The results obtained via empirical formulas, engineering measurements, physical analog model, and numerical simulation were compared. Taking a conservative stance, numerical simulation results would prevail when retaining coal (rock) pillars. Thus, for the 1101 working face of the Baituyao Coal Mine, the heights of the caving zone and the water-conducting fracture zone were 25 and 280 m, respectively, and the fracturing-to-mining height ratio was 14.0. Then, overburden failure mainly occurred at the study area's two ends of the open-off cut. Vertical fissures and bed-separated fissures were periodically developed and closed. As the working face advanced continuously, the height of the water-conducting fracture zone also increased. In the meantime, the ground surface collapsed substantially. The fractured zone finally had a trapezoidal shape, tapering from the bottom to the top. At last, the Neogene and Jurassic overburden strata in the study area feature a short diagenetic history, a low degree of cementation, and ease of softening in the presence of water. Due to the weakly cemented overburden and the Neogene weak aquifer, the risk of water and sand bursts exists in this working face under high-intensity mining.

**Author Contributions:** Conceptualization, K.C.; methodology, K.C.; software, Y.G.; validation, K.C. and Y.G.; formal analysis, L.C.; investigation, Z.L. and Q.Z.; resources, K.C.; data curation, K.C.; writing—original draft preparation, K.C.; writing—review and editing, K.C.; visualization, K.C.; supervision, K.C.; project administration, K.C.; funding acquisition, K.C. All authors have read and agreed to the published version of the manuscript.

**Funding:** This research was funded by the National Natural Science Foundation of Xinjiang Uygur Autonomous Region of China, grant number 2017D01C067 and 2021B03004-1. This research was funded by the second batch of National Natural Science Foundation of Xinjiang Uygur Autonomous Region of China in 2022 (Study on chain mechanism of geological hazards caused by repeated mining in multi coal seams at the northern foot of Tianshan mountains). This research was also funded by Open Fund Project of State Key Laboratory for Geomechanics and Deep Underground Engineering, China University of Mining and Technology, grant number SKLGDUEK2119.

**Institutional Review Board Statement:** Ethical review and approval were waived for this study due to the studies not involving humans or animals.

**Informed Consent Statement:** Patient consent was waived due to the studies not involving humans.

**Data Availability Statement:** The data used to support the findings of this study are included within the manuscript.

**Conflicts of Interest:** The authors declare no conflict of interest.

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
