# Peer review of "Overburden Failure Associated with Slicing Mining in a Super Thick Coal Seam under Special Weak Aquifers"

_water, doi:10.3390/w14233882_

Round 1
Author Response
Dear reviewer,
Thank you very much for your comments and suggestions.
Those comments are all valuable and very helpful for revising and improving our paper, as well as the important guiding significance to our researches. We have studied comments carefully and have made correction which we hope meet with approval. Revised portion are marked in red in the paper. The main corrections in the paper and the responds to the reviewer’s comments are as flowing:
Responds to the reviewer’s comments:
Point 1: The geological situation shows multi aquifers with unconfined as well as confined aquifers. The geological conditions need to be re-visited. If the aquifer is confined how can it be directly recharted with atmospheric precipitation? This could only be possible if the aquiclude is not continuous.
Response 1: The study area is a coal bed mainly filled with fissure water. Through the actual control of the main minable coal seams in the study area by pumping test, it is judged that the stratum of Xishanyao Formation in the middle Jurassic system is a direct water filled aquifer, and the stratum of Dushanzi Formation in the Neogene Pliocene system is an indirect water filled aquifer. Surface water is an important source of recharge for the Quaternary aquifer and the Neogene Dushanzi Formation aquifer. It supplies the Jurassic aquifer through the Quaternary aquifer and the Neogene aquifer. Therefore, there is a close hydraulic connection between surface water and groundwater.
Point 2: The simulation only considers elastic modulus, poisson ratio cohesion and friction angle? How have the input parameters for the model be derived? What are the boundary conditions? Overall, there is a lack of references especially for the geological conditions and lithology.
Response 2:(1) The four input parameters including elastic modulus, Poisson's ratio, cohesion and friction angle can satisfy the calculation of numerical simulation.
(2) Before establishing the numerical simulation model, it is necessary to generalize the geological conditions of the study area. The displacement of the left, right and bottom boundary of the numerical model is constrained, and the upper part is a free boundary. At the same time, in order to reduce the boundary effect, 120 m coal pillars are reserved on both sides.
Point 3:The differences and similarities between physical numerical and empirical model should be explained better such as the very big difference in the height of the water flowing fracture zone?
Response 3: The main reason for the difference in the height of the water conducting fracture zone obtained from the physical simulation and numerical simulation analysis is that the physical simulation only simulates the 10m thick coal seam due to the limitation of the test conditions; The coal seam with a thickness of 20m is mined through numerical simulation. Therefore, the results differ greatly.
Point 4:There is a lack of description on the mining process. How is the mining actually being done and how is the water being handled during mining?
Response 4: The mining parameters of the mine are supplemented in 2.1 Geological Conditions.
Point 5:Why would the aquifer be considered as specially weak and not just weak ? Why would the coal seam be classified as super thick and not just thick ?
Response 5: (1)Compared with the mining area in the east of China, the groundwater in the west mining area is weak in water yield, and the overlying rock has a weak cementation and diagenetic time period. As a result, water inrush accidents will still occur under this special condition of weak water yield. For example, on May 12, 2009, the roof water inrush accident occurred in the process of mining the lower coal seam 1 at the working face 1106 on the east wing of the Okobluk mine in Kuqa County, Xinjiang, China, with the maximum water inrush reaching 1335 m3/h, As a result, the working face was flooded, causing huge economic losses.
(2)The Coal Geology of China divides the coal seams with a single layer thickness of more than 8 m into very thick coal seams (Yang Qi et al., 1979). At present, the thickness of minable coal seams with fully mechanized top coal caving technology has reached about 20 m (Kang Tianhe et al., 2007). In this paper, from the perspective of underground coal mining, and in combination with the current mining technical conditions, the maximum thickness of coal seams that can be mined by fully mechanized top coal caving mining method at one time is about 20 m, which is called huge thick coal seams (Xu Mengtang, 2014).
Point 6:Line 44: unfavorable water-retaining conditions, it is not clear hear what unfavorable water-retaining conditions mean here.
Response 6: In the face of the fragile ecological environment in western China, it is necessary to consider the water conservation mining scheme in the mining area.
Point 7:Line 60: Revise: “caving mining” do you mean block caving method ?
Response 7: “caving mining”mean “sub-level caving”method.
Point 8:Lines 64 to 67: This part links better with paragraph 1 a the beginning.
Response 8: The review of research results and innovation points of this study are added.
Point 9:Line 68: “water-conserving mining”, better revise to water conserving mining operations
Response 9: “water-conserving mining” revise to “water conserving mining operations“.
Point 10:Line 77: “slicing mining” should be slice mining, see also line 81 and line 89
Response 10: “slicing mining” revise to “slice mining”, see also line 81 and line 89.
Point 11: Line 83: “the aquifers have poor water-retaining properties.” Please explain why the aquifers have poor water retaining properties
Response 11:According to the pumping test results of the pore fissure weak water rich aquifer (IV) of the Middle Jurassic Xishanyao Formation, the unit water inflow of borehole ZK10-1 is 0.0075l/s · m; The unit water inflow of borehole ZK7-2 is 0.0013l/s · m; The unit water inflow of borehole ZK5-1 is 0.0007l/s · m, indicating that the water yield of this aquifer is weak.
Point 12: Line 97 “Gobi landforms” Please explain, are you referring here to the Gobi desert ?
Response 12:“Gobi landforms” revise to “Gobi desert”.
Point 13: Line 101: “only reverse fault” should be with only the reverse fault…
Response 13:“only reverse fault” revise to “the reverse fault”.
Point 14: Line 104: “The significant parts” should be: Significant parts….
Response 14:“The significant parts” revise to “ Significant parts”.
Point 15: Table 1: Water retaining property. How is this defined or has this been defined ? What does “/” mean does it mean “Not available ?” Why is there no water retaining property for the quaternary layer ? Why not for formation VI ?
Response 15:(1)“Water retaining property“revise to “water abundance”。
(2)The Quaternary system is in unconformable contact with the underlying strata. The aquifer is recharged from atmospheric rainfall and snow melt water. Because the thickness of the aquifer is 0~5m, it is evaporated after recharge.
(3)The burnt rock fissure phreatic aquifer is formed due to the deformation of the rock caused by the fire and baking of coal seams, the fissure development, and the recharge of atmospheric precipitation and snowmelt water.
Point 16: Fig 2: Revise “Aquifier” Revise “Pore Aquifer” also “Pore Fissure Aquifer”. How does the figure title relate to the figure ?
Response 16:(1)“Pore Aquifer” revise to “Porous aquifer of Pliocene Dushanzi Formation”,“Pore Fissure Aquifer” revise to “Aquiclude of the Middle Jurassic Toutunhe Formation”.(2)revise to “Fig 2. Hydrogeological profile of the study area “
Point 17:Line 120: The coal seam has a shallower burial depth near the…. Shallower than what ?
Response 17:near the coal seam outcrops.
Point 18:Line 122: As a result, the groundwater in the Neogene aquifer recharges the pore-fissure water retaining aquifer. Please explain how the groundwater in the Neogene aquifer recharges the fissure aquifer and why the shallower coal seem burial depth is important for the recharging of the aquifer
Response 18:Based on the analysis of hydrogeological conditions in the mining area, the stratum of Xishanyao Formation of Middle Jurassic System is a direct water filled aquifer, and the stratum of Dushanzi Formation of Neogene Pliocene System is an indirect water filled aquifer.
Point 19:Line 124: the fissure confined aquifer composed of burnt rocks is recharged by atmospheric precipitation. If this is a confined aquifer it would not be directly recharged by atmospheric precipitation. This should be revised.
Response 19:“ confined aquifer” revise to “ unconfined aquifer”.
Point 20:Line 125: which then becomes confined water in the fissures of burnt rocks. This does not sound plausible. Please explain how the fissured aquifer is connected to the coal seam.
Response 20:“ confined aquifer” revise to “ unconfined aquifer”.The burnt rock stratum is a special fissure aquifer, which is located at the coal seam outcrop.
Point 21:Line 127 giving rise to hydraulic connections between aquifers. If this is the case one of the aquifers is not confined. Are there any previous data such pumping tests available to confirm your hydrogeological model ?
Response 21:According to the pumping test in the mining area, the hydraulic connection between the aquifer and the aquiclude is determined.
Point 22:Line 131: The overall overburden thickness is insignificant.. Please explain why it is insignificant, wouldn’t the overburden thickness be also important when it comes to subsidence e.g. ?
Response 22:“ insignificant” revise to “small”.
Point 23:Line 133: include Neogene and Jurassic conglomerate… Are the authors still referring to the Quaternary deposits here ?
Response 23:It does not refer to Quaternary sediments here.
Point 24:Fig. 3 Lithology; Are there references or reports for the lithology, how has the lithology been determined ?
Response 24:The formation lithology is obtained from the collection of the borehole histogram of the mine.
Point 25:Table 2: lithology and mechanical properties: How have the mechanical properties been determined ?
Response 25:It is determined according to the laboratory test, including uniaxial compressive strength and direct shear test.
Point 26:In Table 2 why do we have three columns with the label “Water Saturated” ? Are these values in percentage, how has it been determined ?
Response 26:When conducting uniaxial compressive strength and direct shear test, the uniaxial compressive strength, cohesion and internal friction angle under dry and saturated conditions will be conducted according to the specification.
Point 27:Table 2 also needs to be reformatted
Response 27:
Point 28:Line 159: Revise “slicing mining”
Response 28:“slicing mining” revise to “slice mining”.
Point 29:Line 161 In the present study, the calculation was conducted under a cumulative mining thickness of 15 m. .. Is this a realistic assumption since the coal thickness exceeds 20 m as stated previously ?
Response 29:However, the empirical formula only applies to the situation where the mining thickness of a single slice is 1-3 m, and the cumulative mining thickness does not exceed 15 m. The thickness of the coal seam in this study area is 20 m, which has certain limitations for the calculation of the overburden failure height of the layered mining of huge thick coal seams. Therefore, this paper mainly calculates the cumulative mining thickness of 15m.
Point 30:Fig,4 : Please explain why the different layers with these exact thicknesses were incorporated for the model.
Response 30:Different strata are generalized according to the drilling histogram in the study area. Therefore, the formation lithology and thickness can be accurately expressed.
Point 31:Fig,4 : Please explain why the different layers with these exact thicknesses were incorporated for the model.
How do the dimensions in Figure 4 relate to Figure 5. Are the dimensions in Figure 4 and 5 and 6 still the same ?
Response 31:(1)Different strata are generalized according to the drilling histogram in the study area. Therefore, the formation lithology and thickness can be accurately expressed.
(2)Figure 4 is an engineering geological model generalized according to stratum lithology and stratum thickness, and Figure 5 is a physical simulation made indoor at a scale of 1:240 on the basis of the engineering geological model.
Point 32:Line 188: numerical program for discontinuous structures… The “structures” in the numerical model seem rather continuous. Is there any other reason why this software has been chosen ?
Response 32:UDEC is a discrete element numerical simulation software, which has the advantages of simulating the complex mechanical behavior of rock mass structure, and can grasp the structural and mechanical characteristics of rock strata from a macro perspective when studying the migration law of overburden.It includes: â‘ On the movement mechanism of discrete blocks, large displacement deformation is allowed for rock blocks, and sliding, rotation, detachment and fracture can be achieved along the joint surface;â‘¡ The core of simulation calculation is the calculation of contact force between blocks. The software can automatically identify new contact types according to different shapes of rock blocks.
Point 33:Fig 7: Is the black layer the coal seam ? Wasn’t the coal seam to dip at an angle ? What do the elements outside of the rectangle in Figure 7 represent ?
Response 33:(1)The black layer is the coal seam.
(2)Because the overburden deformation and damage caused by the mining of huge thick coal seam is much stronger than that of thin coal seam, the physical simulation test may not be successful if the dip angle factor is considered again. The research team will also continue to explore the deformation and failure characteristics of overlying rock in the extremely thick coal seam under the dip angle.
(3)These symbols outside represent the control of boundary conditions. The displacement of the left, right and bottom boundary of the numerical model is constrained, and the upper part is a free boundary.
Point 34:Fig 7: The mining scheme and parameters in the numerical simulation were consistent with those of the physical simulation. Please explain how it was ensured that the parameters in the physical simulation were exactly the same as the numerical simulation ?
Response 34:On the basis of generalizing the engineering geological model, ensure the consistency of formation lithology, formation thickness, mechanical parameters, mining speed, etc.
Point 35: Table 3: Mechanical parameters… What are the mechanical parameters based on. Where do they come from ? What is the unit for Elastic modulus ?
Response 35:(1)The mechanical parameters are mainly obtained through laboratory tests.
(2)The unit of Elastic modulus is MPa.
Point 36:What are the boundaries of the numerical model ?
Response 36:The displacement of the left, right and bottom boundary of the numerical model is constrained, and the upper part is a free boundary. At the same time, in order to reduce the boundary effect, 120 m coal pillars are reserved on both sides.
Point 37:Line 220: in the dry and saturated states, respectively… How was this simulated ?
Response 37:When conducting uniaxial compressive strength and direct shear test, the uniaxial compressive strength, cohesion and internal friction angle under dry and saturated conditions will be conducted according to the specification. Therefore, when using empirical formula to calculate, the mechanical parameters under saturated state are taken for calculation according to the most unfavorable situation.
Point 38:Line 242: gradually formed as mining proceeded in the working face… How was this achieved in the analogue model
Response 38:In the physical analog model, the coal seam is replaced by small wooden strips. In the process of simulating the coal seam mining, the method of pulling wooden strips can be used to simulate the coal seam mining.
Point 39:Line 282: The difference between the results from physical and numerical simulations was 15.6 m. What does the value of 15.6 m represent ? Please explain.
Response 39:15.6m represents the development height of the water conducting fracture zone.
Point 40:Figure 9: It is not clear where the first slice in the figure is or where the second slice is.
Response 40: The red area refers to the first layer and the yellow refers to the second layer.
Point 41:Line 334: I suggest the delete the word “scope”
Response 41: “scope” revise to “region”.
Point 42:Line 360: Please outline the “Water-flowing fractured zone: in the physical model and the numerical model, can it be visualized in the figures ?
Response 42: It can be visualized in the figures.In the physical simulation, the fractures formed in the mining overburden are water conducting fracture zones, and in the numerical simulation, the red lines are water conducting fracture zones.
Point 43:Table 4: How can the very big differences in height of the water flowing fracture zone be explained across physical model, empirical formula and numerical model ?
Response 43: The applicable scope of the formula in the "three lower coal mining" specification is when the single layer mining thickness is between 1~3m and the cumulative mining thickness is not more than 15m;Due to the limitation of test conditions, physical simulation only simulates the mining of 10m coal seam thickness;The mining of 20 m coal seam thickness is simulated numerically. Therefore, the results differ greatly.
Point 44:Line 385: “simple geological conditions” Based on the description the geological conditions are everything else but simple
Response 44: Simple geological conditions refer to stratum lithology, geological structure, hydrogeological conditions, etc
Point 45:Line 406 “ease of softening in the presence of water” This has been mentioned before, where does the “ease of softening in the presence of water” come from ? What is the reason ?
Response 45: Because the lithology of Jurassic and Cretaceous strata in the mining area of western China is characterized by short diagenetic time, low cementation and easy softening when encountering water. It is easy to soften when encountering water because there are clay minerals in the lithology of Jurassic and Cretaceous strata.
Point 46:Line 421 Taking a conservative stance, numerical simulation results would prevail when retaining coal (rock) pillars. Please explain why numerical simulations would prevail with coal (rock) pillars.
Response 46: Because the width of coal (rock) pillars is related to the height of water conducting fracture zone, the higher the height of water conducting fracture zone, the greater the width of coal column.Therefore, from the perspective of unfavorable engineering, the width of coal (rock) pillars are designed based on the calculation results of numerical simulation.
Best wishes,
Kai CHEN

Reviewer 2 Report
Reviewer Comments
Paper title: Overburden Failure Associated with Slicing Mining in a Super Thick Coal Seam Under Special Weak Aquifers
The present manuscript describes combined empirical calculations with engineering analogy, physical simulation, and numerical simulation to comprehensively analyze the characteristics of mining-induced overburden failure. The study showed that the overburden in the study area had several unfavorable engineering geological characteristics, including ease of softening in the presence of water.
A manuscript has a practical application and also provides important theoretical for the next studies.
The paper can be accepted for publication after providing the corrections mentioned below.
Point 1. The abstract section sounds unclear. It is presented like description of the results. The abstract should follow the MDPI style of structured abstracts:
- Background (place the question addressed in a broad context and highlight the purpose of the study);
- Methods (describe briefly the main methods);
- Results (summarize the article's main findings);
- Conclusion (indicate the main conclusions or interpretations).
Point 2. Keywords need to be modified. Please use words not combinations of words or phrases.
Point 3. In the Introduction section, an enhanced literature review is required. For this study, the authors have used only 26 literature sources. It seems insufficient for such type of research. It will be great if the authors show some description in context – Why it is important to conduct this study? Impact of water on the daily rock subsidence and mining closure must be analyzed to show the complexity of the research.
Point 3. Can the expected result be used or implemented within other coal basins or hydro-geological conditions? If yes, then how? What limitations?
Point 4. The aim and the tasks must be highlighted at the end of the Introduction section.
Point 5. What software was selected for numerical simulation. Is it FLAC? If yes, what version?
Point 6. What are boundary conditions for the numerical model?
Point 7. The novelty of the paper must be highlighted in the conclusions section.
Point 8. Please provide a short description of further research.
Point 9. You should delete (1), (2), and (3) in the conclusion section
Point 10. There are papers that I have reviewed in the past years. Please consider the suggested research in your paper when enhancing the literature review. I believe they are worth considering in your paper.
Rudakov, D., & Westermann, S. (2021). Analytical modeling of mine water rebound: Three case studies in closed hard-coal mines in Germany. Mining of Mineral Deposits, 15(3), 22-30. https://doi.org/10.33271/mining15.03.022
Bazaluk, O., Sadovenko, I., Zahrytsenko, A., Saik, P., Lozynskyi, V., & Dychkovskyi, R. (2021). Forecasting Underground Water Dynamics within the Technogenic Environment of a Mine Field: Case Study. Sustainability, 13(13), 7161. https://doi.org/10.3390/su13137161
Rudakov, D., & Inkin, O. (2022). Evaluation of heat supply with maintaining a safe mine water level during operation of open geothermal systems in post-coalmining areas. Mining of Mineral Deposits, 16(1), 24-31. https://doi.org/10.33271/mining16.01.024
Point 11. In general, I must admit that a very good study was performed, and I will recommend your paper for publication after careful revision.
Author Response
Dear reviewer,
Thank you very much for your comments and suggestions.
Those comments are all valuable and very helpful for revising and improving our paper, as well as the important guiding significance to our researches. We have studied comments carefully and have made correction which we hope meet with approval. Revised portion are marked in red in the paper. The main corrections in the paper and the responds to the reviewer’s comments are as flowing:
Responds to the reviewer’s comments:
Point 1: The abstract section sounds unclear. It is presented like description of the results. The abstract should follow the MDPI style of structured abstracts:
- Background (place the question addressed in a broad context and highlight the purpose of the study);
- Methods (describe briefly the main methods);
- Results (summarize the article's main findings);
- Conclusion (indicate the main conclusions or interpretations).
Response 1: Revised portion are marked in red in the paper.
Point 2: Keywords need to be modified. Please use words not combinations of words or phrases.
Response 2: The keyword are modified as super thick coal seam; overburden failure; physical simulation; numerical simulation.
Point 3:In the Introduction section, an enhanced literature review is required. For this study, the authors have used only 26 literature sources. It seems insufficient for such type of research. It will be great if the authors show some description in context – Why it is important to conduct this study? Impact of water on the daily rock subsidence and mining closure must be analyzed to show the complexity of the research.
Response 3: According to experts' opinions, the importance of carrying out this research work has been increased, and the recommended references have been added.
Point 4:Can the expected result be used or implemented within other coal basins or hydro-geological conditions? If yes, then how? What limitations?
Response 4: The study area is located in the southern margin of Junggar Basin, Xinjiang Uygur Autonomous Region, and belongs to the southern Junggar coalfield. The geological conditions and mining parameters are representative. If they are applied to other mining areas in the coalfield, they can be analyzed by engineering geological analogy. In addition, the follow-up research team will continue to promote the research results in the region.
Point 5:The aim and the tasks must be highlighted at the end of the Introduction section.
Response 5: Revised portion are marked in red in the paper.
Point 6:What software was selected for numerical simulation. Is it FLAC? If yes, what version?
Response 6: We used UDEC software, version 5.0.
Point 7:What are boundary conditions for the numerical model?
Response 7: The displacement of the left, right and bottom boundary of the numerical model is constrained, and the upper part is a free boundary. At the same time, in order to reduce the boundary effect, 120 m coal pillars are reserved on both sides.
Point 8:The novelty of the paper must be highlighted in the conclusions section.
Response 8: According to the expert opinions, the conclusion is revised, which reflects the innovation of this research work. The coal mining conditions and hydrogeological structure characteristics of the mining areas in western China are quite different from those in eastern China. In the mining area in eastern China, the characteristics of mining overburden failure are very different from those in the mining area in western China, mainly showing a saddle type.
Point 9:Please provide a short description of further research.
Response 9: The mining of huge thick coal seams in western China will aggravate the surface damage. The next research plan will use InSAR technology, unmanned aerial vehicle remote sensing technology and surface monitoring and other means to study the rules and mechanisms of surface damage under the mining conditions of huge thick coal seams.
Point 10:You should delete (1), (2), and (3) in the conclusion section
Response 10: According to expert opinions, (1), (2), and (3) in the conclusion are deleted.
Point 11: There are papers that I have reviewed in the past years. Please consider the suggested research in your paper when enhancing the literature review. I believe they are worth considering in your paper.
Response 11:Thank you for reviewer's comments.I have added your suggested references in the introduction.
Point 12: In general, I must admit that a very good study was performed, and I will recommend your paper for publication after careful revision.
Response 12:Thanks to the comments and suggestions of the reviewers, the research team will continue to carry out in-depth research on coal mining exacerbating surface damage.
Best wishes,
Kai CHEN
Round 2
Reviewer 1 Report
There is still a lack of references for geological hydrogeological conditions.
Many questions raised in the previous review received a response in the reply to author comments but were not addressed in the text.
Title under Citation and in Manuscript is different
Line 44: Grammar
Line 59: "safety problems of fully mechanized sub-level caving of ultrathick coal seams under reservoirs" Please explain what is meant by "fully mechanized sub-level caving" Please also define explain the term "caving zone" which is used later.
Line 64: Past Tense
Line 87: check font, delete "however"
Line 98: Define "Ultrathick"
Line 113: Correct : "The study area has hilly and Gobi desert"....
Table 1: Give references for this information. Is it from this study ? Where does the information come from ?
Line 134: Please explain how a shallower burial depth can be damaging to an impermeable layer.
Line 137: Correct to "fissured"
Line 139: Please define "burnt rocks" are the authors referring to coal here ?
Fig: 3: Give reference for this information. Is it from this study ? Where does this information come from?
Table 2: Give reference for this information. Is it from this study ? Where does this information come from?
Line 182: "A physical analog model is one of the commonly used methods to obtain the height of the water-flowing fractured zone in the overburden": Please explain how a physical analog method can be a common method to obtain the height of the "water-flowing fractured zone".
Line 192: the strength was 360: Please provide unit.
Line 194: "One 120 m coal pillar was retained on each model side to reduce boundary effects" Please explain how the coal pillar on each side of the model would reduce boundary effects. Was the model really that large 120 m ? Was it 120 m tall or wide ?
Line 198: Define "open-off cut"
Line 199: "and the working face was advanced by 10 cm at each mining step" Please explain how the working face was advanced in the model.
Fig. 7: I cannot see the coal pillars on each model side that supposed to reduce boundary effects. Are they missing in the figure ?
Line 220: "The mining scheme and parameters in the numerical simulation were consistent with those of the physical simulation." Please explain how it was ensured that the parameters in the numerical simulation are exactly the same as in the physical model (e.g. friction angle, Elastic Modulus..)
Line 230: was 36.93 and 7.94 MPa: Which formula was used for the calculation ?
Table 4: Please explain the difference in fracture zone between physical and numerical model
Line 392: " In addition, other techniques" Please explain which other techniques should be introduced and why.
Line 397: "simple geological conditions" Please explain why the geological conditions and the lithology shown on figure 3 can be considered simple.
Fig. 11 and Fig 12 are very poor. There is no scale bar. There are no units. It is not clear what the colous are supposed to represent. The colors in legend mostly do not appear in the figure.
Fig. 13: Is this figure from this study ? It should be moved much further upfront into the results and possibly even further up in the manuscript to explain failure characteristics to the reader.
Line 413: " the water-flowing fractured zone presented a trapezoidal shape" Can any of the result figures be referred to here or is this an assumption.
Line 422: The risk of water and sand burst lingers when the overburden is damaged abnormally.: Please explain what is meant by "damaged abnormally".
Conclusions: Are there no recommendations from this study ? e.g. to avoid overburden failure.
Conclusions: The coal mining conditions and hydrogeological structure characteristics of the mining areas in western China are quite different from those in eastern China: Why do the authors compare in the conclusions coal mining and hydrogeological conditions in western and eastern China ? This has not been part of the present study at all.
Author Response
Dear expert,
Thank you very much for your comments and suggestions.
Those comments are all valuable and very helpful for revising and improving our paper, as well as the important guiding significance to our researches. We have studied comments carefully and have made correction which we hope meet with approval. Revised portion are marked in red in the paper. The main corrections in the paper and the responds to the reviewer’s comments are as flowing:
Responds to the reviewer’s comments:
Point 1: Line 44: Grammar
Response 1: It was modified according to the expert opinions.
Point 2: Line 59: "safety problems of fully mechanized sub-level caving of ultrathick coal seams under reservoirs" Please explain what is meant by "fully mechanized sub-level caving" Please also define explain the term "caving zone" which is used later.
Response 2: Fully mechanized sub-level caving is the abbreviation of comprehensive mechanized coal mining in coal mines, which generally refers to the mining method with a mechanization rate of more than 95%. It is a long wall coal mining method comprehensive mining technology, involving well mining, and is suitable for mining medium thick coal seams with stable coal seams, hard roof, no fault and gentle dip and thick coal seams in layers by long wall method.
The caving zone refers to the part of rock stratum where the overlying rock mass of the coal seam is completely collapsed due to the mining of the working face. The rock in this layer is characterized by irregularity, crushing swelling and poor compactness.
Point 3:Line 64: Past Tense
Response 3: It was modified according to the expert opinions.
Point 4:Line 87: check font, delete "however"
Response 4: According to expert opinions,"however" is deleted.
Point 5:Line 98: Define "Ultrathick"
Response 5: "Ultrathick" means "super thick".
Point 6:Line 113: Correct : "The study area has hilly and Gobi desert"....
Response 6: Revised portion are marked in red in the paper.
Point 7:Table 1: Give references for this information. Is it from this study ? Where does the information come from ?
Response 7: Because this paper is a study of the overburden failure law of the weak aquifer in western China, and the classification of hydrogeological structure type is the basis of this study. Therefore, it is necessary to analyze the hydrogeological structure type of the mining area under study. The data in this table are from the data collected in the mining area.
Point 8:Line 134: Please explain how a shallower burial depth can be damaging to an impermeable layer.
Response 8: Because in the coal seam outcrop area, the upper part is missing the water resisting layer, which will lead to atmospheric rainfall and snow melt water directly entering the mining face.
Point 9:Line 137: Correct to "fissured"
Response 9: It was modified according to the expert opinions.
Point 10:Line 139: Please define "burnt rocks" are the authors referring to coal here ?
Response 10: The burnt rock is a special kind of rock formed from surrounding rock baked by coal seam spontaneous combustion.The burnt rock is not a coal seam, it is located above the coal seam.
Point 11: Fig: 3: Give reference for this information. Is it from this study ? Where does this information come from?
Response 11:Borehole histogram is the basis for subsequent physical simulation and numerical simulation of formation lithology modeling. The chart data is derived from the data collected in the mining area.
Point 12: Table 2: Give reference for this information. Is it from this study ? Where does this information come from?
Response 12:Because in the process of empirical formula, physical simulation and numerical simulation, the physical and mechanical parameters of roof rock are required. Therefore, it is necessary to analyze the lithology and mechanical parameters of the roof.
Point 14: Line 182: "A physical analog model is one of the commonly used methods to obtain the height of the water-flowing fractured zone in the overburden": Please explain how a physical analog method can be a common method to obtain the height of the "water-flowing fractured zone".
Response 14:“water-flowing fractured zone".” revise to “ water-conducting fracture zone”.As shown in the figure below.
Point 15: Line 192: the strength was 360: Please provide unit.
Response 15: 360 kPa.
Point 16: Line 194: "One 120 m coal pillar was retained on each model side to reduce boundary effects" Please explain how the coal pillar on each side of the model would reduce boundary effects. Was the model really that large 120 m ? Was it 120 m tall or wide ?
Response 16:120m
Point 17:Line 198: Define "open-off cut"
Response 17:Open-off cut refers to driving along the starting line of the coal face, driving a roadway along the coal seam between the transport roadway and the air return roadway to form an independent air return system. The mining face can be arranged only after the air return system is formed
Point 18:Line 199: "and the working face was advanced by 10 cm at each mining step" Please explain how the working face was advanced in the model.
Response 18:In the physical analog model, the coal seam is replaced by small wooden strips. In the process of simulating the coal seam mining, the method of pulling wooden strips can be used to simulate the coal seam mining.The height and width of a small wooden strip are 1cm. According to the mining plan, 10 small wooden strips are sampled each time, which is 10cm.
Point 19:Fig. 7: I cannot see the coal pillars on each model side that supposed to reduce boundary effects. Are they missing in the figure ?
Response 19:The area in the red box in the figure below is the coal pillar.
Point 20:Line 220: "The mining scheme and parameters in the numerical simulation were consistent with those of the physical simulation." Please explain how it was ensured that the parameters in the numerical simulation are exactly the same as in the physical model (e.g. friction angle, Elastic Modulus..)
Response 20:The data in Figure 3 and Table 2 are the basis of physical and numerical simulation. The difference is that in the physical simulation test, a certain proportion will be reduced based on the real physical and mechanical parameters according to the similarity comparison. In the numerical simulation, the real physical and mechanical parameters are used.
Point 21:Line 230: was 36.93 and 7.94 MPa: Which formula was used for the calculation ?
Response 21:The average uniaxial compressive strength of roof rocks of coal seams B1 and B2 in the saturated state was taken as 12.93 MPa for the calculation.The calculated formula is in line 184.
Point 22:Table 4: Please explain the difference in fracture zone between physical and numerical model
Response 22:Because of the test conditions, the physical simulation only conducted the mining of the 10m thick coal seam; The numerical simulation has done the mining of 20 m thick coal seam. Although the mining thickness is different, the deformation and failure characteristics of overburden are relatively consistent under the mining condition of 10m thick coal seam.
Point 23:Line 392: " In addition, other techniques" Please explain which other techniques should be introduced and why.
Response 23:In addition to the physical simulation and numerical simulation applied in the paper, field tests can also be used to determine the height of the water conducting fracture zone, such as borehole television method.
Point 24:Line 397: "simple geological conditions" Please explain why the geological conditions and the lithology shown on figure 3 can be considered simple.
Response 24:Compared with the mining area in eastern China, the geological structure, formation lithology and hydrogeological conditions of the mining area studied are relatively simple, which are manifested in simple geological structure, single formation lithology and weak water yield of the aquifer.
Point 25:Fig. 11 and Fig 12 are very poor. There is no scale bar. There are no units. It is not clear what the colous are supposed to represent. The colors in legend mostly do not appear in the figure.
Response 25:The vertical displacement diagram and vertical stress diagram have been revised according to expert opinions. In addition, the colors in the legend are displayed in the figure, but the displayed area is small and not obvious.
Point 26:Fig. 13: Is this figure from this study ? It should be moved much further upfront into the results and possibly even further up in the manuscript to explain failure characteristics to the reader.
Response 26:This figure focuses on the analysis of the failure characteristics of overburden under the mining of huge thick coal seams obtained through numerical simulation, and deeply analyzes the formation process of various fractures.
Point 27:Line 413: " the water-flowing fractured zone presented a trapezoidal shape" Can any of the result figures be referred to here or is this an assumption.
Response 27:It can be seen from the overburden failure characteristics of physical simulation and numerical simulation that the overburden failure area is trapezoidal distribution as a whole.As shown in the figure below.
Point 28:Line 422: The risk of water and sand burst lingers when the overburden is damaged abnormally.: Please explain what is meant by "damaged abnormally".
Response 28:Because the overburden structure characteristics and hydrogeological structure characteristics of mining areas in the east and west of China are very different, on the one hand, if the height of the water flowing fracture zone calculated according to the empirical formula specified in the specification is smaller; On the other hand, the overburden failure law and characteristics are very different from those of the eastern mining areas in China. Therefore, according to the empirical formula of mining areas in eastern China, water and sand inrush disasters will not occur, but because of the special overburden structure and hydrogeological structure of mining areas in western China, overburden damage and water and sand inrush disasters are abnormal.
Point 29:Conclusions: Are there no recommendations from this study ? e.g. to avoid overburden failure.
Response 29:According to the empirical formula, physical analog model and numerical simulation, the height of water-conducting fracture zone formed by mining and the deformation and failure law of overburden are suggested.
Point 29:Conclusions: The coal mining conditions and hydrogeological structure characteristics of the mining areas in western China are quite different from those in eastern China: Why do the authors compare in the conclusions coal mining and hydrogeological conditions in western and eastern China ? This has not been part of the present study at all.
Response 29:According to the experts' suggestions, this part of conclusions has been revised.
Best wishes,
Kai CHEN

Reviewer 2 Report
Dear authors,
I am satisfied with the corrections provided by you.
This study is an important contribution to sustainable mining.
Congratulations to the authors.
Author Response
Dear profess,
Thank you very much for your comments and suggestions.
Those comments are all valuable and very helpful for revising and improving our paper, as well as the important guiding significance to our researches. Due to the mining of huge thick coal seams in western China will aggravate the surface damage. The next research plan will use InSAR technology, unmanned aerial vehicle remote sensing technology and surface monitoring and other means to study the rules and mechanisms of surface damage under the mining conditions of huge thick coal seams.Finally, thank you for your comments and suggestions.
Best wishes,
Kai CHEN
